# Unsupervised learning of categorical structure

**Matteo Alleman**
Department of Neurobiology and Behavior
Columbia University
New York City, NY 10027
ma3811@columbia.edu

**Stefano Fusi**
Department of Neurobiology and Behavior
Columbia University
New York City, NY 10027

## Abstract

Humans occasionally reason using logic and abstract categories, and yet most state of the art neural models use continuous distributed representations. These representations are impressive in their learning capabilities, but have proven difficult to interpret, or to compare to biological representations. But continuous representations can sometimes be interpreted symbolically, and a distributed code can seem to be constructed by composing abstract categories. We ask whether it is possible to detect and get back this structure, and we answer that it sort of is. The demixing problem is equivalent to factorizing the data into a continuous and a binary part: $\mathbf{X} = \mathbf{W}\mathbf{S}^T$. After establishing some general facts and intuitions, we present two algorithms which work on low-rank or full-rank data, assess their reliability on extensive simulated data, and use them to interpret neural word embeddings where we expect some compositional structure. We hope this problem is interesting and that our simple algorithms provide a promising direction for solving it.

## 1 Introduction

In biological and artificial learning systems, compositional structure is important to flexible behavior, yet difficult to detect at the representational level. Neural representations are rarely factorized into purely-selective concept neurons; when there is a neat conceptual structure it is most often embedded into high-dimensional neural modes (Kaufman et al., 2022; Higgins et al., 2021; She et al., 2021; Bernardi et al., 2020). Machine learning systems which either explicitly incorporate abstraction, or have just found to exhibit it, also use distributed continuous representations which are rarely factorized (Altabaa et al., 2024; Rigotti et al., 2022; Mikolov et al., 2013).

This poses a challenge for both mechanistic interpretability and structural alignment between representations. While there has been impressive success recently using sparse autoencoders for interpretability (Huben et al., 2024) the fact that items vary in degree of feature membership makes it hard to treat the discovered features as logical or fully compositional concepts. It could also be nice to have the tidiness of a discrete structure for the purpose of comparing representations. Unstructured representational alignment is somewhat ill-defined (Ahlert et al., 2024), and so another method for finding the 'content' of a representation could be a useful addition to the toolbox.

We will show that the discovery of latent categories can be formulated as a binary matrix factorization. Given a representation matrix, $\mathbf{X}$, we will try to find a binary representation, $\mathbf{S}$, which encodes an assignment of items to logical variables. We think of these logical variables as abstract 'concepts', which are optimized to match the geometry of the data. Many structures–including disentanglement, clustering, hierarchy, linear ordering, and hybrids of these–can be captured by this simple model. But finding an optimal factorization is a difficult combinatorial problem, and has not been extensively studied in the general case.

Preprint.

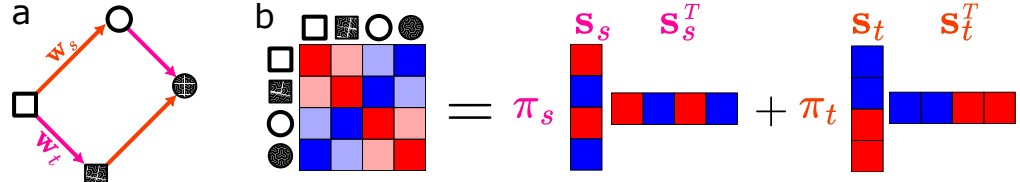

Figure 1: Illustration of a binary decomposition of a kernel matrix. (**a**) A representation with compositional structure in which items are made by adding shared component vectors. (**b**) The resulting (linear) kernel matrix, which we model as a conic combination of rank-one binary matrices.

## 1.1 Existing algorithms

Like many matrix factorizations, our method has an aesthetic similarity to the eigenvalue decomposition and PCA. In fact, when the concepts are uncorrelated with each other, i.e. they are orthogonal after subtracting the means, then it is actually equivalent to PCA. However–the rarity of this situation aside–PCA is not likely to find binary features when there is some approximate rotation-invariance to the solution, like with the square. Furthermore, when there are multiple factorizations, the sparsest one will have correlated features in general and thus not be recoverable.

We can also view our problem as a case of non-disjoint similarity-based clustering. Dasgupta (2016) defined an objective similar to ours for hierarchical clustering, and many of the theoretical results therein could be of interest. Work has also been done on generalizing $k$-means clustering to allow for overlap (Cleuziou, 2007; Whang et al., 2015) which is conceptually similar but uses a different cost function with a different implied generative model. The mixed-membership stochastic blockmodel (Airoldi et al., 2008) is another famous clustering model tackling a similar problem, and the connection to ours is made explicit by Sørensen et al. (2022).

In the community detection and applied math literature, our factorization problem has been studied as (semi) binary matrix factorization (SBMF). Remarkably, in special cases an algebraic solution is available via tensor decomposition (Sørensen et al., 2021), but it is highly sensitive to violations of its assumptions. There are several optimization-based approaches (Zhang et al., 2007; Kolomvakis and Gillis, 2023; Sørensen et al., 2022) which are generally built around the assumption of very low-rank data, and thus may not be applicable in the general case.

The specific formulation of our model closely follows the Binary Component Decomposition of Kueng and Tropp (2021). The algorithm they present (and the noise-robust algorithm of Kolomvakis and Gillis (2023)) requires that the category vectors satisfy a type of general position property called 'Schur independence' (Laurent and Poljak, 1996). If there are sufficiently few categories (fewer than approximately $\sqrt{2p}$ for $p$ items) then this is not a restrictive requirement – yet as we will see it excludes many interesting categorical structures like hierarchies. One of our main contributions on top of that work is to show that mild regularization sometimes allows for the recovery of meaningful structure in this problem despite lack of identifiability.

## 2 Problem formulation

### 2.1 Model

In the kind of conceptualisation we seek it should be possible to get from one thing to another by switching certain concepts. For example, in Fig. 1a we can get from the white square to the shaded square by adding the $t$ vector; or we can predict what we will see when we add both the $s$ and the $t$ vector. This kind of conceptual navigation is what we will try to discover from the data.

In matrix terms, we are modelling the data as the product of a continuous and a $\{0, 1\}$-valued matrix:

$$\mathbf{X} \sim \mathbf{W}\mathbf{S}^T$$

in which the columns of $\mathbf{X} \in \mathbb{R}^{n \times p}$ are the representations of data points, the columns of $\mathbf{W} \in \mathbb{R}^{n \times b}$ are the representations of pure categories (henceforth 'weight vectors'), and the columns of $\mathbf{S} \in \{0, 1\}^{p \times b}$ indicate which points belongs to a given category (henceforth 'concept vectors'). We

will denote individual concept vectors by $\mathbf{s}$. Note that we will work with an affine model, which can be achieved by an explicit intercept term or by including a trivial concept of all 1.

Constrained matrix factorization is a challenging problem, especially when some factors are discrete. To make things somewhat easier, we will assume orthogonal weights, $\mathbf{W}$. This helps us because our objective function (see next section) depends only on the Gram matrices, so it is possible to fit the concepts independently of the weights. Notice that the Gram matrix of the model prediction, $\hat{\mathbf{X}}$, depends only on the category vectors and the squared norms of the components:

$$\hat{\mathbf{X}}^T \hat{\mathbf{X}} = \mathbf{S} \mathbf{W}^T \mathbf{W} \mathbf{S}^T = \mathbf{S} \mathbf{\Pi} \mathbf{S}^T$$

where $\mathbf{\Pi}$ is a (positive) diagonal matrix containing the squared norms of the weights. This is implicitly the approach taken by Kueng and Tropp (2019) and Kolomvakis and Gillis (2023) for semi-BMF. From this perspective, we are modeling the dot product between each pair of observations by the weighted sum of the number of shared concepts:

$$\mathbf{x}_i^T \mathbf{x}_j \sim \sum_{\alpha=1}^{b} \pi_\alpha \mathbf{S}_{i,\alpha} \mathbf{S}_{j,\alpha} \tag{1}$$

which is represented visually in Fig.1c.

### 2.1.1 Existence and uniqueness

Without an orthogonality constraint, clearly all data, $\mathbf{X}$, can be decomposed with an exact SBMF; just set $\mathbf{W} = \mathbf{X}$ and $\mathbf{S} = \mathbb{I}$. With orthogonal $\mathbf{W}$, not all $\mathbf{X}$ can be factorized exactly, and it is NP-hard to check for a particular $\mathbf{X}$ (Deza and Laurent, 1997). Nevertheless, most data is fairly 'close' to an exactly-embeddable representation (Laurent and Poljak, 1996), and we give some tiny examples in Figure 2. Among our examples, the square, tetrahedron, and tree are exactly embeddable, while the grid and hexagon are best approximations (as defined in the following section).

Exact or approximate, the optima are rarely unique[1]. A necessary and sufficient condition for uniqueness would be NP-hard to check (Deza and Laurent, 1997), but several sufficient conditions have been derived of varying restrictiveness and complexity (Kueng and Tropp, 2021; Sørensen et al., 2022). A crude intuition: the higher-dimensional the data, the more possible solutions. Two extreme examples are the $b$-cube and the $p$-simplex. A hypercube has $p$ points in $b$ dimensions, and its binary representation is unique; meanwhile, a $p$-simplex (every point equidistant) has $p - 1$ dimensions, and a tremendous number of possible representations, including the identity matrix and all Hadamard matrices. Because of this, we argue for a sparsity inductive bias in Section 2.2.

### 2.1.2 Graphical representation

High-dimensional binary vectors might be hard to interpret, and we would like a visualization tool. Hierarchical clustering is much more useful when visualized with a dendrogram, a tree on which observations are leaves, and cluster assignments can be recovered by cutting at a certain depth of the tree. Just as our problem generalizes hierarchical clustering, we can generalize the dendrogram.

Given a set of $b$ concepts, $\mathbf{S}$, we can isometrically embed each point (row of $\mathbf{S}$) on a 'partial cube', i.e. an isometric subgraph of the $b$-bit hypercube. In this representation (e.g. Fig. 2, bottom row), nodes are connected by an edge when they differ by one concept. Some hidden nodes might be necessary to form the graph. Concepts can be read out by cutting across 'parallel' edges (those corresponding to the same concept), and for visualization it is often easier to color the edges and make them directed by fixing one arbitrary point as the 'origin' (as we have done in Fig. 2). By analogy to a dendrogram, which is a type of partial cube, we will call this kind of graph an 'analogram'.

Every partial cube has an easily-obtained unique binary representation (Deza and Laurent (1997), and see the Appendix A.1 for explicit construction), but there are a combinatorial number of graphs whose binary representation is $\mathbf{S}$. The analogram should be the smallest such graph. Unfortunately, like almost everything related to this problem, finding such an analogram is NP-complete and it needn't even be unique (Knauer and Nisse, 2019). Nevertheless, we made a heuristic algorithm that works well on moderately-sized $\mathbf{S}$, and always on trees (see A.1). For larger and non-tree structures, even a minimal graph will be too messy to view, so other visualization tools would be required.

---

[1]By unique, we mean up to sign/bit-flip symmetry, since these are trivial symmetries of our objective.

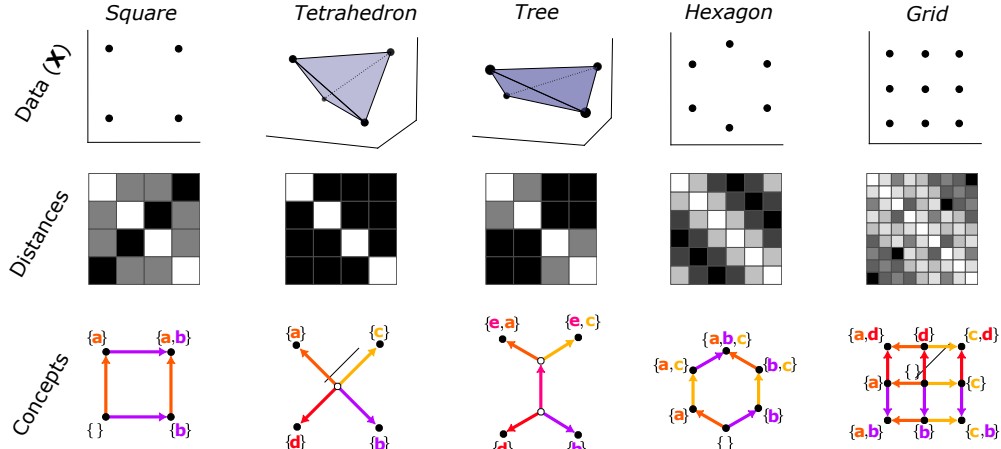

Figure 2: Examples of some categorical structures recoverable from geometry. The concepts are found by optimizing Equation 3 with brute force (i.e. setting $\mathbf{S} = \{0, 1\}^p$) and node regularization. The graphs (see Section 2.1.2) are drawn by manual inspection. Notice that the concept labels (in curly brackets) can be gotten by cutting the graph at the corresponding edges, and labelling the partition on the arrow side. We chose the source (no concept) nodes arbitrarily.

## 2.2 Objective

Since an exact fit is not always possible (or even desirable), we must define goodness of approximation. The centered kernel alignment (CKA) is commonly used to measure representational similarity in machine learning and neuroscience. It has an interesting interpretation as a non-parametric measure of dependence (Gretton et al., 2005; Sejdinovic et al., 2013) but for our purposes is just an empirically useful objective. Unlike the mean squared error between the kernels, it is translation and scale invariant, which is desirable in a measure of geometric similarity. Compared to the MSE between $\mathbf{WS}^T$ and $\mathbf{X}$, the CKA formulation also has fewer free parameters since we avoid fitting $\mathbf{W}$ (due to assumed orthogonality).

The CKA is the cosine similarity of the flattened, double-centered Gram matrices. Specifically, if we have $p \times p$ Gram matrices $\mathbf{K} = \mathbf{X}^T\mathbf{X}$, and $\mathbf{Q} = \mathbf{Y}^T\mathbf{Y}$, then double-centering means computing the dot products after subtracting the mean from each dimension. If we can define the centering matrix $\mathbf{H} = \mathbb{I}_p - \frac{1}{p}\mathbf{1}_p$, then feature centering is expressed by $\bar{\mathbf{K}} = \mathbf{H}\mathbf{X}^T\mathbf{X}\mathbf{H}$. For short, we will denote the Frobenius inner product applied to double-centered matrices by $\langle \mathbf{HKH}, \mathbf{HQH} \rangle_F = \langle \mathbf{K}, \mathbf{Q} \rangle_{\mathbf{H}}$.

So, the problem we want to solve is

$$\underset{\mathbf{Q}=\mathbf{S\Pi S}^T}{\arg\max} \quad \frac{\langle \mathbf{K}, \mathbf{Q} \rangle_{\mathbf{H}}}{\sqrt{\langle \mathbf{K}, \mathbf{K} \rangle_{\mathbf{H}} \langle \mathbf{Q}, \mathbf{Q} \rangle_{\mathbf{H}}}} \tag{2}$$

In this form the CKA is not nice to optimize, but since our constraining set is a cone[2] it is equivalent to the Frobenius distance between the double-centered matrices. In our case this reduces to:

$$\underset{\mathbf{S}\in\{0,1\}^{p\times b}, \boldsymbol{\pi}\in\mathbb{R}_+^b}{\arg\min} \quad \boldsymbol{\pi}^T \left(\bar{\mathbf{S}}^T\bar{\mathbf{S}}\right)^2 \boldsymbol{\pi} - 2\,\mathbf{1}^T \left(\mathbf{X}^T\bar{\mathbf{S}}\right)^2 \boldsymbol{\pi} \tag{3}$$

where $\cdot^2$ is element-wise squaring and $\bar{\mathbf{S}} = \mathbf{HS}$. If we fix $\mathbf{S}$, this is just a non-negative least squares problem in $\boldsymbol{\pi}$ and we can expect at most $\binom{p}{2}$ non-zero elements. For sufficiently small $p$ this can be used to solve the problem by enumerating over all possible $\mathbf{s}$ vectors and keeping those with non-zero $\boldsymbol{\pi}$ value. In general, though, we use it to refine a solution which might have more concepts than strictly necessary, or to derive an iterative optimizer.

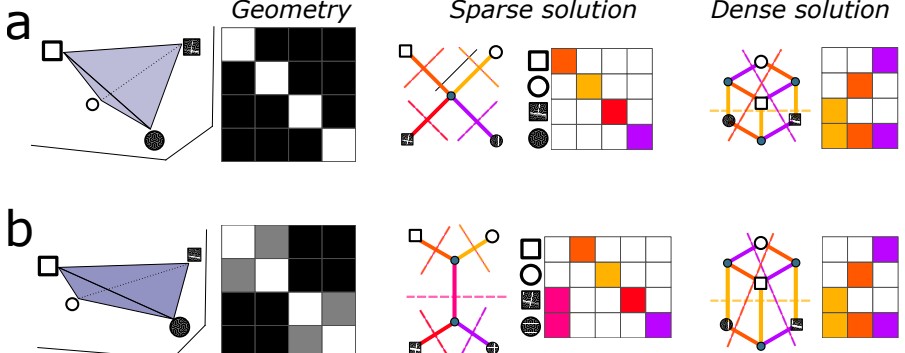

Figure 3: Examples of degenerate solutions.

### 2.2.1 Regularization

There could be an astronomical number of optima for a given geometry, so we need an inductive bias to break the tie. Parsimony is desirable, but there are several ways to define it. Using the minimal number of concepts might seem like a reasonable choice – we will argue that this is not the right measure of complexity, and instead we should try to minimize the size of the analogram (Section 2.1.2). Since that is very hard to do directly, we propose regularizing for sparsity.

Consider the two examples in Figure 3. There are two possible embeddings for the 4-simplex (panel a), one with 4 concepts but only 1 hidden node, and one with 3 concepts but three hidden nodes. A similar situation is true for the two solutions of a stretched simplex (panel b), and in this case the structure with more concepts is potentially more in line with our intuition. But as was mentioned, we cannot optimize for this directly. Instead, we notice that the simpler graphs tend to have less intersection between the concepts, as well as sparser concepts. For that reason, we will try to encourage sparsity as a proxy for graph complexity.

To encourage sparsity, we can just add a linear term to the loss: if $\mathbf{r}_\alpha = \min\{\mathbf{1}^T \mathbf{s}_\alpha, p - \mathbf{1}^T \mathbf{s}_\alpha\}$, we add the term $\mathbf{r}^T \boldsymbol{\pi}$ to Equation 3 to be minimized. We have considered other proxy which better correlate with graph complexity, but do not include them here .

## 3 Optimization

Being a challenging combinatorial problem, we cannot expect efficient solutions that work in every situation. There are already remarkably effective approaches for very low-rank data, but they do not always fail gracefully when their assumptions are violated. We will therefore adapt some of these existing techniques to the non-identifiable case, by iteratively sampling category vectors. In addition, we develop a complementary online algorithm which builds all category vectors simultaneously, but one item at a time. Note that both algorithms can in principle work on any positive semidefinite matrix, and are thus quite general. Each approach has its strengths, and also ample room for improvement.

### 3.1 Rejection sampling for low-rank data

When the dimensionality of the data is sufficiently small, Kueng and Tropp (2021) and Kolomvakis and Gillis (2023) provide efficient algorithms based on randomly sampling columns of $\mathbf{S}$. Their methods are based on the following observation: If the data admits a factorization of the form $\mathbf{X} = \mathbf{W}\mathbf{S}^T$, and $\mathbf{W} \in \mathbb{R}^{n \times b}$ has full column rank, then, since the linear mapping can be inverted, each column of $\mathbf{S}$ is also in the rowspace[3] of $\mathbf{X}$. So, these algorithms randomly search for vectors $\mathbf{s} \in \{-1, 1\}^p$ that are in the rowspace of $\mathbf{X}$. When the rank, $r$, is very small, it is possible to use exhaustive search since there are at most $2^r$ such binary vectors (Slawski et al., 2013).

---

[2]i.e. Closed under non-negative linear combinations.

[3]i.e. the image of $\mathbf{X}^T$

The aforementioned algorithms are able to search efficiently by assuming that the $\mathbf{s}$ vectors are Schur-independent. In that case, because of the special structure of the set of correlation matrices (Laurent and Poljak, 1996; Kueng and Tropp, 2021), each $\mathbf{s}$ can be found via semidefinite programming (SDP). But in the general case the SDP is not guaranteed to have a rank-one optimum, and would require a heuristic rounding step. In such a situation it might not be worth the computational burden of solving an SDP, and so we propose a first-order method based on Hopfield networks.

In the presence of noise, the true concept vectors might not be exactly in the rowspace of $\mathbf{X}$, but just be the closest among other binary vectors. If $\mathbf{U}$ are the right singular vectors (with non-zero singular values) of $\mathbf{X}$, then we expect the true concept vectors, $\mathbf{s}$, to be local maxima of $E(\mathbf{s}) = \mathbf{s}^T \mathbf{U} \mathbf{U}^T \mathbf{s}$. This is precisely (the negative of) the energy function of a Hopfield network, which we can maximize by iteratively updating $\mathbf{s}_t \leftarrow \text{sign}(\mathbf{U} \mathbf{U}^T \mathbf{s}_{t-1})$ from some initial guess.

There is an interesting connectionist interpretation of this procedure. Let us imagine the concept $\mathbf{s}$ is the binary response pattern of neuron to the whitened inputs – then the Hopfield updates amount to Hebbian plasticity. If the weights of the neuron are $\mathbf{m}$, then the update above tells us that it should be set to $\mathbf{m}_i = \sum_j \mathbf{U}_{i,j} \mathbf{s}_j$, which is a Hebbian rule. Taking this interpretation, we call this algorithm the binary autoencoder[4] (BAE). It works by randomly sampling $\mathbf{s}$ vectors according to Algorithm 1, with a tolerance parameter $\epsilon$ and inverse temperature $\beta$ on an exponential annealing schedule.

We find that this simple algorithm performs very well on simulated low-rank data. For $p$ points, we draw $\sqrt{2p}$ random $\mathbf{s}$ vectors, which almost certainly admit a unique decomposition, and randomly embed them in $d$-dimensions. We then add iid Gaussian noise to achieve a specified signal-noise ratio (SNR). Since the algorithm scales with the rank of $\mathbf{X}$, which is $\leq \sqrt{2p}$, we observe a nearly linear scaling with $p$ (Fig. 4a). Furthermore, we recover the ground-truth factors for most SNR values up to the largest we tried: $p = 2^{12}$. For SNR of 1 (log SNR of 0) we can recover up until around 100 points, at which point the errors sharply increase (Fig. 4b). It is likely that there are similar inflection points for higher SNRs at larger values of $p$, but we did not explore that far.

---

**Algorithm 1** Rejection sampler for BAE

1: **function** SAMPLE($\mathbf{U} \in \mathbb{R}^{p \times n}, \epsilon > 0$)
2: $\quad \mathbf{m} \sim \mathcal{N}(\mathbf{0}_n, \mathbb{I}_n)$
3: $\quad \mathbf{b} = \mathbf{0}$
4: $\quad$ **while** not converged **do**
5: $\quad\quad \mathbf{s}_t \sim \text{Bernoulli}(\sigma[\beta(\mathbf{U}\mathbf{m} + \mathbf{b})])$
6: $\quad\quad \mathbf{m} \leftarrow \mathbf{U}^T \mathbf{s}_t$
7: $\quad\quad \mathbf{b} \leftarrow \frac{1}{p}\mathbf{1}\mathbf{1}^T \mathbf{s}_t$
8: $\quad$ **if** $\frac{1}{p}\|\mathbf{U}^T \mathbf{s}_t\|_2^2 \geq 1 - \epsilon$ **then**
9: $\quad\quad$ **Return** $\mathbf{s}_t$
10: $\quad$ **else**
11: $\quad\quad$ **Return** SAMPLE($\mathbf{P}, \epsilon$)

---

**Algorithm 2** Iterative item assignment

1: **function** FIT($\mathbf{X}, K \in \mathbb{N}, n \leq p$)
2: $\quad$ **for** $k = 1, .., K$ **do** $\quad \triangleright$ *Parallel branches*
3: $\quad\quad \mathbf{S}(k) = \{0, 1\}^n$
4: $\quad\quad \boldsymbol{\pi}(k) = \arg\min_{\boldsymbol{\pi}} 3$
5: $\quad\quad$ **for** $i = n, ..., N$ **do**
6: $\quad\quad\quad \varepsilon \sim \mathcal{N}(0, \mathbb{I})$
7: $\quad\quad\quad \hat{\mathbf{p}}, \pi_0 = \arg\min 4 + \varepsilon^T \hat{\mathbf{p}}$
8: $\quad\quad\quad \mathbf{S}(k) \leftarrow \begin{pmatrix} \mathbf{S} & \mathbf{S} & \mathbf{0}_n \\ \mathbf{1}_b^T & \mathbf{0}_b^T & 1 \end{pmatrix}$
9: $\quad\quad\quad \boldsymbol{\pi}(k) \leftarrow (\hat{\mathbf{p}}, \boldsymbol{\pi} - \hat{\mathbf{p}}, \pi_0)$
10: $\quad$ **Return** $\mathbf{S}, \boldsymbol{\pi}$

---

### 3.2 Iterative refinement for full-rank data

When $\mathbf{X}$ is full rank, we cannot use Algorithm 1. It is possible to run it the top few principle components of $\mathbf{X}$, and this often will recover some high-level concepts, but it will not recover hierarchical structures which are genuinely full rank. In such a situation, every concept is in $\text{Im}(\mathbf{X}^T)$, so we must optimize our objective function (3) over $\{0, 1\}^p \times \mathbb{R}_+^p$. Our approach will be to update the item-wise category assignments to match similarities to other items.

Let us assume we have a solution for the first $n$ items. When we receive a new item, our data kernel, $\mathbf{K}$, and model kernel $\mathbf{Q}$ will each get a new column (and row) and a new diagonal element:

$$\mathbf{K}^{(n+1)} = \begin{pmatrix} \mathbf{K}^{(n)} & \mathbf{k} \\ \mathbf{k}^T & k_0 \end{pmatrix}, \quad \mathbf{Q}^{(n+1)} = \begin{pmatrix} \mathbf{Q}^{(n)} & \mathbf{q} \\ \mathbf{q}^T & q_0 \end{pmatrix}$$

---

[4]This name begs a question: why not use a literal autoencoder? A tanh network trained to maximize $E(\tanh(\mathbf{U}\mathbf{m}))$ with gradient descent can work, but it is substantially slower to converge than the fully binary algorithm and is more prone to bad optima. When a reconstruction loss is added, we struggled to make it work at all. Nevertheless, it is a direction we are interested in pursuing.

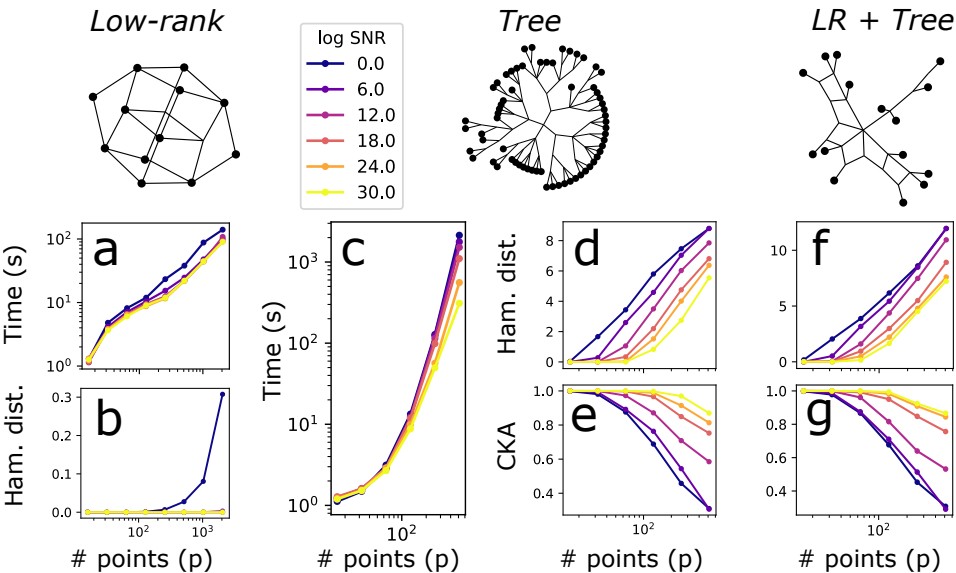

Figure 4: Numerical experiments. All were carried out on 2.9 Ghz, 16-core CPUs without parallelization, and plots are the average of 25 random seeds. (**a**) The time taken by Algorithm 1. (**b**) Average Hamming distance from ground-truth concept to nearest discovered concept. Values are normalize by $p$ to be able to see the slight non-zero values in some lines. (**c**) Time taken by Algorithm 2. (**d**, **f**) Raw (i.e. unnormalized) Hamming distances from ground truth to nearest discovered concept, for random tree-structured and hybrid data. (**e**, **g**) CKA of the model fit to $\mathbf{X}$.

Our algorithm will modify $\mathbf{S}$ and $\boldsymbol{\pi}$ in order maximize the objective, only changing $\mathbf{q}$ and $q_0$. To motivate our approach, we will use the neural network interpretation of the BAE. Imagine a subpopulation of neurons which all respond in the same way to the first $n$ items. represented by column $\alpha$ of $\mathbf{S}$ (with associated $\pi_\alpha$). In response to item $n + 1$, only a certain fraction, $p_\alpha$, might respond. At the same time, another sub-population which has not responded to any previous item might also chime in. If we define the response probabilities as $\mathbf{p} = (p_1, ..., p_b)$, and the strength of the newly-active population as $\pi_0$, then the effect on the total kernel is:

$$\mathbf{q} = \mathbf{S}\boldsymbol{\Pi}\mathbf{p}, \quad q_0 = \boldsymbol{\pi}^T\mathbf{p} + \pi_0$$

Finding the optimal $\mathbf{p}$ and $\pi_0$ is a quadratic problem. We have to account for centering, which is shown in the Appendix A.2, but the end result is a $(d + 1)$-dimensional interval-constrained QP:

$$\underset{\hat{\mathbf{p}}, \pi_0}{\arg\min} \quad 2\left\|\bar{\mathbf{S}}(\hat{\mathbf{p}} - \langle\hat{\mathbf{s}}\rangle) - \mathbf{k}\right\|_2^2 + \frac{n}{n + 1}\left((\mathbf{1} - 2\langle\mathbf{s}\rangle)^T\hat{\mathbf{p}} + \pi_0 + \langle\hat{\mathbf{s}}\rangle^T\langle\mathbf{s}\rangle - k_0\right)^2 \quad (4)$$

$$\text{s.t.} \quad \mathbf{0} \le \hat{\mathbf{p}} \le \boldsymbol{\pi}, \ 0 \le \pi_0$$

where we have defined $\langle\mathbf{s}\rangle = \frac{1}{p}\mathbf{S}^T\mathbf{1}$, $\langle\hat{\mathbf{s}}\rangle = \boldsymbol{\Pi}\langle\mathbf{s}\rangle$ and $\hat{\mathbf{p}} = \boldsymbol{\Pi}\mathbf{p}$. We can adapt our complexity regularization to work iteratively as well, while preserving convexity. Having done this, we split any columns with fractional $\mathbf{p}$ and delete any categories with a $\boldsymbol{\pi}$ value less than some small threshold. The general outline is provided in Algorithm 2.

The iterative assignment algorithm is greedy and sensitive to item order. In order to deal with resulting local minima, we add some randomization and can run multiple fits in parallel (the $\varepsilon$ term in Algorithm 2). In principle, the intuition we used to motivate the algorithm can be made more concrete by implementing it with an explicit population, where the optimization of Equation 4 is implemented by recurrent weights. It would be interesting to explore such a model, but we find that it is faster and more stable to solve the problem directly by convex optimization (using the cvxpy package, Diamond and Boyd (2016)).

We test this algorithm on hierarchical concepts, as well as a hybrid low-rank and hierarchical structure. To generate this data, we recursively partition the points by randomly choosing a number of partitions between 2 and 4, defining a concept for each partition, and then repeating within each

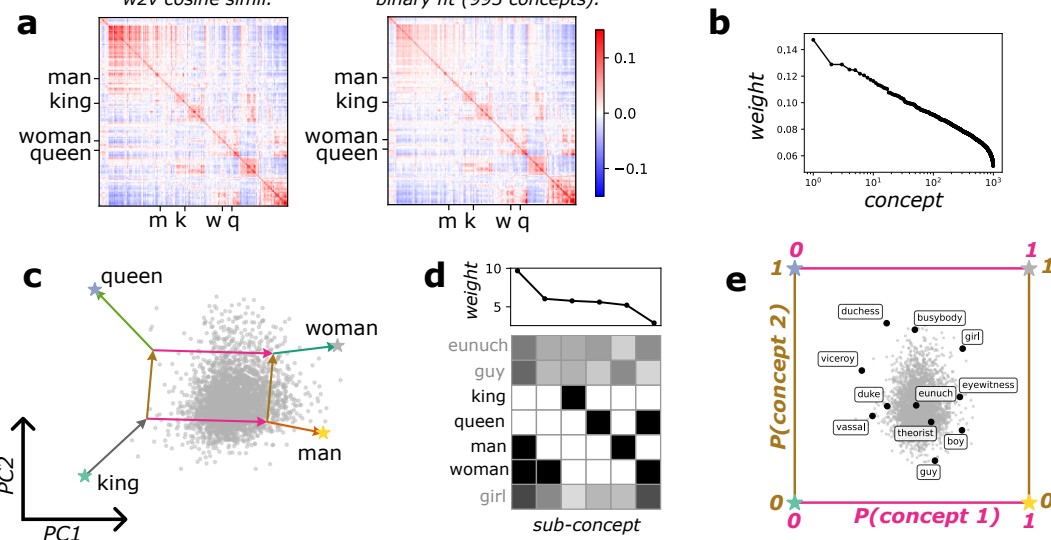

Figure 5: (**a**) Cosine similarity of 3841 words, in the word2vec representation (left) and the binary reconstruction (right). Words were selected using WordNet by taking all hyponyms of 'person' which were also present in word2vec. (**b**) Weight ($\pi$) value of each concept. (**c**) PCA plot of four words of interest, with analogram overlaid and projection of all other selected words. (**d**) Example of sub-concept vectors, which are averages over concept vectors conditioned some subset of points. (**e**) The 'conceptual projection' of all fitted words onto the two high-level concepts. Highlighted words were selected by manual inspection.

partition. The same noise model was used as before. The hybrid structure is just the concatenation of randomly-sampled concepts and the hierarchical ones. Because the number of concepts is much higher, and we are solving a $p \times b$ NNLS on each iteration, the scaling is much poorer than the BAE (Fig. 4 c). We note that, while the complexity appears super-polynomial, this is because of a change in the slope around $p > 64$, which is visible when sampling with finer granularity.

Regularizing for sparsity allows us to recover the ground truth structure up to around 100 points, but since these structures do not have unique solutions, noise very quickly diminishes our ability to recover the ground-truth solution in both cases (Fig. 4d,f). Interestingly, the ground-truth recovery is correlated with our objective (Fig. 4e,g), which suggests that improving performance might also improve recovery of the ground truth.

Finally, we will see how this works on a simple example from word2vec (Mikolov et al., 2013), a well-known word embedding. The embedding is generated by predicting word-context pairs, in which contexts are the neighboring words in the Google News corpus. The embeddings can themselves be seen as a type of matrix factorization (Levy and Goldberg, 2014). What earned word2vec its notoriety is the fact that certain abstract concepts seem to be encoded along approximately parallel dimensions–exactly the kind of structure that we are looking for.

Since word2vec has embeddings for several million words, we need to select a subset to study. WordNet (Fellbaum, 1998) contains manually-encoded 'is-a' relations (i.e. 'dog' is-a 'canine') over a large vocabulary, and we used it to generate a list of words that are descendants of 'person'. This comes out to 3841 words. This is just a way to generate a reasonably-sized dataset which might be somewhat related, and we do not use the hypernymy after this. With around 4000 points in 300 dimensions, this is a good use case for Algorithm 1; we achieve a CKA of around 0.93 (Fig. 5a) using 993 concepts (Fig.5b)

It is hard to make sense of high-dimensional clusters, and the analogram of this embedding will be too dense to be useful, so we will use a small example to orient ourselves. In particular, we will look at the famous 'king:queen::man:woman' quadruplet. To do so, we will take the binary embedding of the four words, call it $\mathbf{S}_{\mathrm{kqmw}}$, and use the unique columns. The analogram of this subset is shown

in Fig. 5c. Like Mikolov et al. (2013), we see a concept which groups together 'king' and 'queen' (concept 1), and one which groups together 'king' and 'man' (concept 2), but we also have word-specific concepts to account for the deviation from a perfect rectangle. In this accounting, concept 1 is more strongly represented than concept 2, as is seen in the correlation matrix.

While these clusters are often interpreted as 'class' and 'gender', it is not clear from just 4 words whether this is the right interpretation. When we take a subset, we are combining multiple dataset-level concepts into a single quadruplet-level concept; that is, there are many dataset-level concepts which group together 'king' and 'queen' vs. 'man' and 'woman'. For example, 'man' and 'woman' are very generic terms, and something like 'specificity' is a concept which could plausibly affect the contexts in which they appear in the news.

When projecting the data onto the top 2 principle components of our four points, it is not clear which concepts are being captured. As we can see in Fig.5c, the quadruplet is not fully captured by just the two balanced features. Instead, we will try to project words in a way that is specific to the concepts we are trying to interrogate. For each word, we can take the average over all concepts conditional on a particular value of the four items of interest, producing a $[0, 1]$ score for each other item (Fig.5d).

In Fig. 5e we are plotting this average value (which is between 0 and 1) for every word in grey, and some specific words are highlighted. In general, the right side of concept 1 seems to be courtly roles rather than rulers *per se*, while the left side are perhaps called 'mundane'. Concept 2 does seem to capture some gendered words ('duke' vs 'duchess' and 'boy' vs 'girl'), and the expected parallelisms are visible in this projection. On the other hand, concept 2 seems less about class than expected – words with low projection are indeed fairly 'kingly', while many words with a high projection relate to crime, like 'victim', 'shooter', and 'eyewitness' (shown in the plot). We note that this qualitative picture persists for many random seeds and hyperparameter choices (annealing schedule, regularization, etc.).

There is clearly much to be desired in the discovered embedding. At least part of that can be addressed by improving the model fit, and we are interested in combining our two algorithms to better fit the more local structure being missed by the low-rank algorithm. On the other hand, some of the counter-intuitive or (not shown) unsavory placements of words in this schema are visible in some form in the raw representations. In that sense, the added interpretability of our method's unambiguous category assignments (along with good estimation of the weight vectors $\mathbf{W}$) could make it useful for removing unwanted concepts present in the dataset.

## 4  Discussion

Here we studied the problem of turning a continuous representation into a logical one. We provided two simple algorithms with complementary use cases and demonstrate their efficacy. When dealing with very low-rank data, we leveraged results from previous work to develop a very fast and fairly robust method based on Hopfield networks. For when the data is higher-dimensional, we developed an online algorithm which iteratively solves convex subproblems. Both models can in principle be implemented by a population of neurons with Hebbian input and recurrent synapses, and though we treat this as a curiosity for now, it raises the question of biological relevance.

To the extent that we, humans, are engaged in concept discovery, our work here could also provide a minimalistic model of that cognitive process. We are not modelling the difficult process of discovering concepts from the real world: our model is linear, and therefore assumes a representation which approximately encodes the relevant concepts. Instead, we can model the process of turning a distributed, noisy code into abstract symbols and structures (Kemp and Tenenbaum, 2008). Future work can address how the inherent challenges of unsupervised abstraction, which we laid out, might be dealt with by more sophisticated algorithms with different inductive biases.

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

# A  Appendix

## A.1  Partial cubes

Here is the sketch of the algorithm for recovering concepts from a partial cube graph: Start with an edge $e = (x, y)$, partition[5] the vertices into those which are closer to $x$ (call them $V_x$) or closer to $y$ (call them $V_y$). Each $V_x$ node gets a 0 label, each $V_y$ node gets a 1. Pick another edge, ignoring from now on any edges which cross the partition, and repeat the process. Here is an illustration of the process for a hexagon graph, coloring edged according to the partitions:

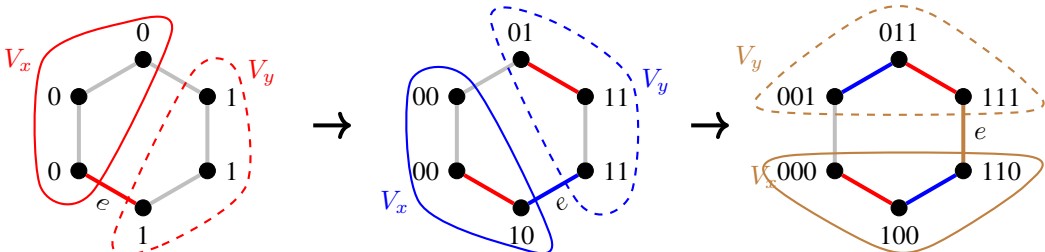

This algorithm is described in several places, but we used chapter 19 of Deza and Laurent (1997).

To go the other way around is hard, as mentioned in the main text. Our heuristic algorithm is based on two components: a global rule for finding hidden nodes, and a local rule for placing edges.

The global rule finds 'necessary' hidden nodes by taking the intersection of all differences between items. We will assume concepts are centered around the first item, such that the first row is all 0. We then take the intersection of all rows. For example, if items 1 and 2 differ by concepts $\{a, b, c\}$, while items 1 and 3 differ by $\{a, b, d\}$, then we will include the hidden node with concepts $\{a, b\}$. This is then repeated after re-centering around all other items.

The local rule is used to add edges to the graph, and ensure it is connected. It is a rule for which bits to add from a given node. Let's again assume we are centered around item 1. Construct a 'disjoint' matrix between concepts, which is equal to 1 only if $\mathbf{s}_\alpha^T \mathbf{s}_\beta = 0$. Then construct a 'superset' matrix, which is 1 only if $\mathbf{s}_\alpha^T \mathbf{s}_\beta = \mathbf{1}^T \mathbf{s}_\alpha$. The rule is that, at a given node $i$, with concepts $\mathbf{S}_i$, we are allowed to add any concepts which (1) are not disjoint with any concepts of $i$ and (2) have all their supersets

---

[5]This rule partitions the graph because partial cubes are bipartite. In fact, the partitioning is the basis of a certain binary relation, the Djokovic-Winkler relation, which is the theoretical basis of this construction.

active in $i$. For example, if I has the concepts ('dog', 'big'), then we could add 'great dane' but not 'pug', and not 'cat'.

By using the global rule for placing hidden nodes, and the local rule for laying down paths between all nodes, we can make a graph that is sometimes of manageable size. It works much better for sparser graphs like trees, and so a more picky method is going to be necessary moving forward.

## A.2   Recursive loss function

Here we show in more detail how to compute the loss for Algorithm 2 upon receiving a new item.

We assume we have a fit for the first $n$ items. The kernels when we see item $n+1$ are:

$$\mathbf{K}^{(n+1)} = \begin{pmatrix} \mathbf{K}^{(n)} & \mathbf{k} \\ \mathbf{k}^T & k_0 \end{pmatrix}, \quad \mathbf{Q}^{(n+1)} = \begin{pmatrix} \mathbf{Q}^{(n)} & \mathbf{q} \\ \mathbf{q}^T & q_0 \end{pmatrix}$$

The loss we want to compute is

$$\left\| \bar{\mathbf{Q}}^{(n+1)}, \bar{\mathbf{K}}^{(n+1)} \right\|_{\mathbf{H}} = \left\langle \bar{\mathbf{Q}}^{(n+1)}, \bar{\mathbf{Q}}^{(n+1)} \right\rangle_{\mathbf{H}} + \left\langle \bar{\mathbf{K}}^{(n+1)}, \bar{\mathbf{K}}^{(n+1)} \right\rangle_{\mathbf{H}} - 2 \left\langle \bar{\mathbf{Q}}^{(n+1)}, \bar{\mathbf{K}}^{(n+1)} \right\rangle_{\mathbf{H}}$$

To compute the required inner products, we need to double-center the matrices. It's easier if we assume that everything is already centered with respect to the first $n$ items. So, let's say that $\bar{\mathbf{k}}$, $\bar{k}_0$, $\bar{\mathbf{q}}$, and $\bar{q}_0$ are the appended elements centered with respect to the first $n$ features[6]. That makes it much simpler to compute the centering with the addition of one new item. We end up with the following update:

$$\left\langle \bar{\mathbf{Q}}^{(n+1)}, \bar{\mathbf{K}}^{(n+1)} \right\rangle_{\mathbf{H}} = \left\langle \bar{\mathbf{Q}}^{(n)}, \bar{\mathbf{K}}^{(n)} \right\rangle_{\mathbf{H}} + 2t\bar{\mathbf{k}}^T\bar{\mathbf{q}} + t^2 \bar{k}_0 \bar{q}_0$$

where I have abbreviated $t = \frac{n}{n+1}$. Plugging these into the loss equation and fiddling a bit, we have:

$$\|\mathbf{K}^{(n+1)} - \mathbf{Q}^{(n+1)}\|_{\mathbf{H}}^2 = \|\mathbf{K}^{(n)} - \mathbf{Q}^{(n)}\|_{\mathbf{H}}^2 + 2t\bar{\mathbf{q}}^T\bar{\mathbf{q}} + t^2\bar{q}_0^2 - 2\left(2t\bar{\mathbf{k}}^T\bar{\mathbf{q}} + t^2\bar{k}_0\bar{q}_0\right) + 2t\bar{\mathbf{k}}^T\bar{\mathbf{k}} + t^2\bar{k}_0^2$$

$$= \|\mathbf{K}^{(n)} - \mathbf{Q}^{(n)}\|_{\mathbf{H}}^2 + 2t \left\|\bar{\mathbf{q}} - \bar{\mathbf{k}}\right\|_2^2 + t^2(\bar{q}_0 - \bar{k}_0)^2 + \text{const.} \quad (5)$$

which we can get by completing the square. This means that we are just solving a (slightly) weighted least squares between the $n$-centered data and prediction. So long as $\bar{\mathbf{q}}$ and $\bar{q}_0$ are linear functions of our parameters (which they will be), minimizing the loss at each step is just a quadratic program.

It remains to write out $\bar{\mathbf{q}}$ and $\bar{q}_0$. The $n$-centered form of our concept update is:

$$\bar{\mathbf{S}} \leftarrow \begin{pmatrix} \bar{\mathbf{S}} & \bar{\mathbf{S}} & \mathbf{0}_n \\ 1 - \langle \mathbf{s} \rangle & -\langle \mathbf{s} \rangle & 1 \end{pmatrix}$$

and remember that the $\boldsymbol{\pi}$ update is

$$\boldsymbol{\pi} \leftarrow (\mathbf{\Pi}\mathbf{p}, \ \mathbf{\Pi}(1 - \mathbf{p}), \ \pi_0)$$

To get the kernel values we need to take the dot product of the last row with the first $n$ rows and with itself (weighting columns by $\boldsymbol{\pi}$). Working through the algebra on that we get:

$$\bar{\mathbf{q}} = \bar{\mathbf{S}}\mathbf{\Pi}(\mathbf{p} - \langle \mathbf{s} \rangle)$$

and

$$\bar{q}_0 = (1 - 2\langle \mathbf{s} \rangle)^T \mathbf{\Pi}\mathbf{p} + \langle \mathbf{s} \rangle^T \mathbf{\Pi}\langle \mathbf{s} \rangle + \pi_0$$

which are indeed linear functions of our parameters, $\mathbf{p}$ and $\pi_0$.

## A.3   Illustrative examples

To build intuition for binary embeddings, we will go over some prototypical cases which can be inspected manually. We will imagine observations coming from three regular latent structures, and ask whether the ground truths are recoverable. In each diagram, the distances given are Euclidean, and the graph distances should match their square.

---

[6]More specifically, if $\mathbf{K}^{(n)} = \mathbf{X}^T\mathbf{X}$, $\langle \mathbf{x} \rangle$ is the average feature of the first $n$ items, and $\mathbf{x}$ is our new item's feature, then $\bar{\mathbf{k}} = (\mathbf{X} - \langle \mathbf{x} \rangle)^T(\mathbf{x} - \langle \mathbf{x} \rangle)$. The same for $\bar{\mathbf{q}}$ and the rest.

### A.3.1 Independent categories

This is the nicest case, one where we observe all possible combinations of the latent categories. The resulting geometry is a $d$-dimensional cube, far lower dimensionality than the number of observations ($2^d$):

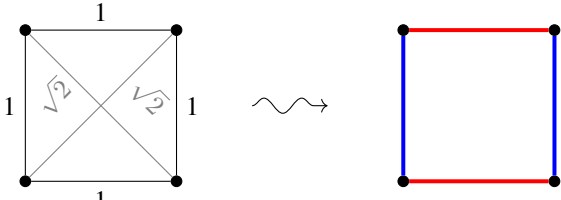

Above, I have drawn an example geometry for four points (on the left), and the corresponding graph of the embedding (on the right). In the graph, each edge is colored according to its corresponding category.

### A.3.2 Mutually exclusive categories

This case is most similar to traditional clustering, in which categories cannot be combined. Every observation belongs to exactly one latent category, and the resulting dimensionality is potentially as high as the number of observations. The resulting geometry is one where every cluster of points is equidistant to every other.

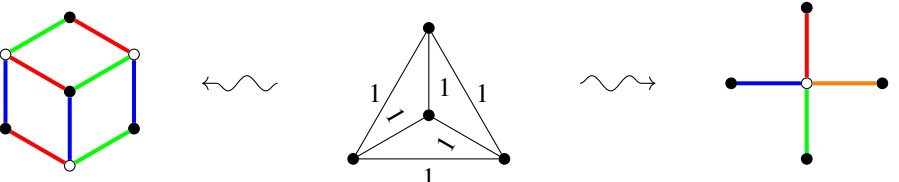

In our 4-item example above, there are two possible embeddings: a 3-dimensional one with more 'hidden' nodes (left) and a 4-dimensional one with fewer hidden nodes (right). The 4d one recapitulates our 'ground truth' latent structure since each item has its own category, but the 3d one does equally well. Intuitively, we might prefer the 4d solution since (1) it does not group together points unnecessarily and (2) there are fewer hidden nodes. Therefore, rather than the embedding dimension as the natural notion of parsimony, we suggest to use the number hidden nodes, or category size.

**Hierarchy**   Instead of a flat clustering, imagine the data fall into hierarchical clusters in which some categories are subsets of others. The resulting geometry has intermediate dimensionality:

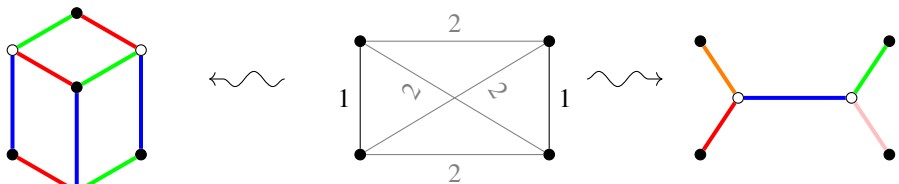

Notice this time that the graph on the left is isomorphic with the one in the previous example, but the blue edges have higher weight (i.e. they are longer in the drawing) to account for the larger distances. Instead, the graph on the right is a tree – something we would expect from hierarchical clustering. One category, the blue edges which separate the two pairs of closest points, appears in both solutions. So, at least some times, some categories might be identifiable even when the embedding as a whole is not.

### A.3.3 An ordinal variable

Our observations now come from one ordered variable – for example, counting. The resulting geometry is a line. While all the previous examples admitted exact embeddings, this geometry is not embeddable on the cube and thus only has an optimal approximation. How can this be expressed in terms of categories?

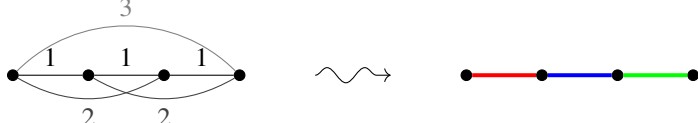

We end up with a graph of the expected topology, but the distances are slightly different since we are trying to match Hamming distances with squared Euclidean distances. With this example we can also see the utility and limits of our model: while we recover something meaningful at the graph level, we need 3 binary variables to describe a 1-dimensional structure[7]. So, while it is possible to get a sensible binary embedding of something fundamentally non-binary, it is not always efficient to do so.

---

[7]It is still possible to see the 1-dimensional structure by computing the 'lattice dimension' of the partial cube graph, which is the minimum dimension of an integer lattice that can contain the graph.

