# OpenReview forum: "Unsupervised Learning of Categorical Structure"
_NeurIPS.cc/2024/Workshop/UniReps — UniReps_

### Official Review · Reviewer_ABih · 2024-10-01
**Evaluation of Latent Category Discovery via Binary Matrix Factorization**

**Rating:** 7
**Confidence:** 1

**Review:**

In this paper, the authors formulate the discovery of latent categories as a binary matrix factorization problem and propose two algorithms (one leveraging previous work, one developed). These algorithms enable a continuous representation to be turned into a logical one.

They demonstrate on simulated data the capacity of each algorithm to recover (until a certain threshold) hierarchical concepts and show successful experimentation on word2vec datasets

### Strengths:
-  Paper is well written
-  Problem formulation is original
-	The advantages and drawbacks of each algorithm are well-discussed
-	Rigorous evaluation of the synthetic data
### Weaknesses:
- Evaluation of several quadruplets would have been relevant

### Comments:
-	Please comment a bit more about why you are sure that you “can do better than CKA 0.8”

---

### Official Review · Reviewer_q87k · 2024-10-03
**Novel direction of work with interesting experiments**

**Rating:** 8
**Confidence:** 3

**Review:**

Summary: The paper addresses the question of whether continuous neural representations can be interpreted discretely as a composition of abstract categories. It also presents two algorithms and uses them to interpret neural word embeddings in this framework.

Strengths:

1. The premise of the work is very interesting and innovative, especially relevant in light of lower interpretability of distributed representations.

2. Older literature is appropriately referenced and the research’s contribution is clearly situated.

3. There are considerations for both low rank and high dimensional data with appropriate solutions.


Weaknesses:

1. The application has been limited to neural embeddings or Word2Vec and it would be interesting to see applications with embeddings of advanced LLMs.

2. The analogue with reasoning capabilities of cognitive science could be developed more.

---

### Official Review · Reviewer_6Rkm · 2024-10-05
**Review for Unsupervised Learning of Categorical Structure**

**Rating:** 8
**Confidence:** 5

**Review:**

This paper proposes a method for extracting discrete categories from continuous representations in neural networks using binary matrix factorization (BMF). The authors introduce two algorithms: one for low-rank data based on a rejection sampler inspired by Hopfield networks, and another for full-rank data using iterative refinement. The method is validated on synthetic data and applied to interpret word2vec embeddings, providing a novel approach for linking symbolic and continuous AI representations.

The approach to extract categorical structure from continuous data using BMF is innovative and addresses a significant gap in neural interpretability. The two proposed algorithms are well-motivated and theoretically grounded. The use of Hopfield-like networks for low-rank data is creative. Besides, the word2vec embedding analysis provides a compelling real-world use case, demonstrating the practical utility of the method. This paper is clearly written with strong mathematical foundations and effective visual aids.

---

### Official Review · Reviewer_PaBG · 2024-10-06
**Interesting matrix-decomposition-based approach towards unsupervised learning of categorical structures**

**Rating:** 4
**Confidence:** 3

**Review:**

This submission proposed that categorical structures can be unsupervised learned using the binary component decomposition of matrices. Section 2.1 explains the problem formulation: Given data, bundle them column-wise to a matrix $X$. Then, seek a binary component decomposition of $X$, finding a composition $W\\cdot S^T$ for $X$ where $S$ has only coefficients 0 and 1. The transfer to learning categorical structure isn’t easy to understand because, in the literature mentioned in the previous introduction section, the matrix $W$ with the weight vectors is composed of vectors that can be interpreted as weights. However, this is different in the application presented here. The wording “pure categories” for the continuous part of the matrix decomposition also leads to more confusion. The model prediction $\\hat{X}$ is not defined. Unfortunately, it is not clear how to seek the model’s prediction. Unfortunately, it is also unclear what a solution for the Gram matrix of the data matrix $X$ has to do with a solution to the initial problem, $X$.
The writing is incomplete in many places, which means that the definitions of variables are missing, and the presented Algorithms lack clarity:
- 2.1 In equation (1), $d=b$?
- 2.1.2 Please revise this section. Give a rigorous definition of what is meant by an “analagram”. It isn’t easy to follow the continuous text. I am not sure whether I grasped the idea correctly. Maybe it is more accessible using Hasse diagrams of finite Boolean algebras instead of hypercube graphs. Aren’t “analograms” just a class of Hasse diagrams?
- 2.2.1 Please clarify what $s_\\alpha$ is.
- 3.1 Algorithm 1 What is $\\sigma, \\beta$? What is $P$ when calling recursion? The condition of the while-loop can not be understood.
- 3.1 Algorithm 2 What is meant by $S(k)=\\{ 0,1 \\}^n$? What’s $\\hat{p}$?

The incomplete algorithms make it impossible to evaluate the writing because it is referenced in the further course.

---

### Decision · Program_Chairs · 2024-10-10

**Decision:**

Accept

**Comment:**

In light of the positive reviewers' feedback and relevancy of the submission, we are pleased to accept this paper for presentation at UniReps 2024. We kindly ask the authors to incorporate the reviewers' suggestions and feedback in the final camera-ready version of the manuscript.